# Research on Mechanical Properties and Influencing Factors of Cement-Graded Crushed Stone Using Vertical Vibration Compaction

**DOI:** 10.3390/ma15062132

**Published:** 2022-03-14

**Authors:** Yingjun Jiang, Huatao Wang, Kejia Yuan, Mingjie Li, Ming Yang, Yong Yi, Jiangtao Fan, Tian Tian

**Affiliations:** 1Key Laboratory for Special Area Highway Engineering of Ministry of Education, Chang’an University, Xi’an 710064, China; jyj@chd.edu.cn (Y.J.); ykj@chd.edu.cn (K.Y.); wsyysnb@163.com (Y.Y.); 2020021070@chd.edu.cn (J.F.); 2019021033@chd.edu.cn (T.T.); 2Henan Provincial Transportation Infrastructure Quality Inspection Station, Zhengzhou 450016, China; 2020221217@chd.edu.cn (M.L.); 2020221219@chd.edu.cn (M.Y.)

**Keywords:** railway engineering, cement-graded crushed stone, mechanical properties, vertical vibration compaction method, strong interlock skeleton dense gradation

## Abstract

To study the mechanical properties of cement-graded crushed stone for use in the transition sections of intercity railways, the growth laws governing unconfined compressive strength, splitting strength and resilience modulus of cement-graded crushed stone and their influencing factors were studied by the vertical vibration compaction method (VVCM). The strength growth equations of cement-graded crushed stone are proposed, and strength prediction equations are established. The research shows the unconfined compressive strength, splitting strength and resilience modulus of cement-graded crushed stone with a strong interlocked skeleton density type (VGM-30) are significantly enhanced to 20, 20 and 17% higher, respectively, than those of standard cement-graded crushed stone. The growth law of mechanical properties of cement-graded crushed stone is similar, with the fastest growth occurring before 14 days, and the rate decreasing after 28 days. The strength growth tended to be stable after 90 days, increasing with the increase in curing time, compaction coefficient and cement dosage. The correlation coefficients (R^2^) of the strength growth prediction models were found to be 0.99, 0.97, and 0.99, respectively. These values can be used to accurately predict the strength growth curve. This paper verifies the superiority of VGM-30 gradation through laboratory tests, providing a reference for gradation selection in the construction of intercity railway transition sections.

## 1. Introduction

A railway foundation bed is one of the necessary components for railway construction. Graded crushed stone is often used in railway subgrade due to its stress dispersion ability. It can effectively reduce the residual deformation of the railway subgrade caused by the dynamic load of trains and improve the stability and bearing capacity [1,2,3,4,5]. With the rapid development of intercity railways worldwide, the requirements for railway subgrades are becoming stricter and more specific. However, the current specifications stipulate multiple, wide-ranging grading types, making the formation of a skeleton structure difficult. The mechanical indicators following on-site compaction are based on the indoor mechanical standard, which cannot predict on-site construction indicators. The heavy compaction method (HCM) and the static compaction method (SCM) are used in the test of graded crushed stone, which may be inconsistent with the actual vibration compaction moulding process and modern heavy traffic [6]. The grading and mechanical indexes stipulated by the specification may not reflect the mechanical properties of the graded crushed stone with accuracy. This inaccuracy results in the limited evaluation and strength prediction of the mechanical properties of the cement-graded macadam and no guiding significance for the actual construction of the graded macadam.

At present, some researchers have carried out studies on cement-graded macadam. Tan et al. formed large-size graded crushed stone specimens by the vibration and rotary compaction method and analysed the effects of gradation parameters on cumulative strain and stability, the test results show that the large-size graded crushed stone has better mechanical properties than the conventional-size graded crushed stone [7]. Long et al. studied the influence of the vibration compaction method and the modified Proctor test on the stress–strain characteristics and shear strength of graded crushed stone, and the results show that the cohesion, friction angle and shear strength of the vibratory compacted sample are 38%, 2° and 10% higher than those of the modified Proctor compacted sample, respectively [8,9]. Yang et al. established a cumulative plastic strain prediction model by studying the change law of material deformation and strength characteristics of heavy railway subgrade graded crushed stone under cyclic loading [10]. Cui et al. studied the mechanical properties of cement-graded crushed stone with a dense skeleton structure, and the results showed that a higher cement content led to better mechanical properties [11]. Chen et al. optimised the parameters of the graded crushed stone transition layer based on mechanical response and strength standards [12]. While Deng et al. discussed the influence of the compaction coefficient, moisture content, and gradation on the mechanical properties of graded crushed stone and established a prediction model for the dynamic resilience modulus [13,14]. Thavathurairaja et al. measured the resilient modulus values of graded crushed stone under various stress states and fitted them into multiple constitutive models to evaluate their applicability [15]. Kam et al. studied the influence of moisture content, per cent fines content, stress, and gradation on the resilience modulus of graded crushed stone and established a model to predict the resilience modulus of graded crushed stone [16]. Lv et al. studied the uniaxial compression, bending and indirect tensile mechanical properties of cement-stabilised macadam; established the strength yield surface of cement-stabilised macadam with increasing loading rates, and comprehensively described the fatigue characteristics of cement-stabilised macadam [17]. Cook et al. evaluated the impact of particle shape and gradation on the durability and mechanical properties of the graded crushed stone used in road structures, and the results showed that the particle shape has an impact on the overall performance [18]. Bilodeau et al.’s study found that the gravel materials exhibited complex elastic–plastic characteristics under wheel load, and the gradation effect was related to the friction between particles. They also emphasised the importance of matrix interlocking and non-matrix aggregate void filling capacity [19].

According to the above studies, cement-graded crushed stone is a subgrade filler with superior performance. However, there is limited research on the grading type and the predictive mechanical strength models of laboratory tests on the graded crushed stone of intercity railway transition sections. Generally, specimen moulding adopts the HCM and the SCM, which cannot accurately reflect the vibration compaction characteristics of on-site engineering, leading to a low correlation between laboratory tests and on-site construction. According to these urgent problems, Jiang et al. developed the vertical vibration compaction method (VVCM) to study the impact of different compaction methods on the performance of road materials. The test showed that the correlation between the specimens by the VVCM and the on-site drilling core sampling is as high as 92%, while the specimen’s mechanical properties and fatigue properties are superior to the general formation methods [20,21,22,23,24,25]. This paper presents a strong interlocked skeleton density-type cement-graded crushed stone using the VVCM moulding method to study the influence of different gradations, curing times, cement dosages and compaction coefficient on the mechanical properties of cement-graded crushed stone. These are compared and analysed. At the same time, according to the strength of the growth equation in predicting the law of strength increase in cement-graded crushed stone, a reliable basis for constructing the transition sections of intercity railways is provided.

## 2. Raw Materials and Test Plans

### 2.1. Raw Material

(1) Graded crushed stone

The main physical properties of graded crushed stone are shown in Table 1. The aggregates used in the test are all from Xi’an, Shaanxi, and meet the requirements of JTG E42-2005 [26].

(2) Cement

The cement used in the test is the P·O42.5 cement produced by Shaanxi Province, the material used in the test meets the requirements of GB 175-2007 [27], and its technical properties are shown in Table 2.

### 2.2. Mixtures

Studies of the mechanical properties and influencing factors of graded crushed stone are planned to select the strong interlocked skeleton density-type cement-graded crushed stone (VGM-30) and the normal graded crushed stone of the transition section. The standard one and two as well as the upper limit (U), median (M), and down limit (D) are compared. See Table 3 [28,29], for the sieve pass rate of each gradation, and see Table 4 for the maximum dry density and optimal moisture content of the selected gradation.

### 2.3. Specimen Preparation Method

In this paper, the vertical vibration compaction method (VVCM) was used to prepare the specimen. The preliminary research and related studies have compared the VVCM with the heavy compaction method (HCM) and static compaction method (SCM), fully demonstrating that the VVCM moulding method has better field correlation than the HCM and SCM. Because of different compaction mechanisms, the mechanical strength of the VVCM-moulded specimens is also higher than those moulded by the HCM and SCM [30,31,32].

(1) Instruments

The overall structure of the vertical vibration test equipment (VVTE) is shown in Figure 1. The equipment is composed of three parts: the control platform, the rotating device and the vibration system. The function of the vibration platform is to adjust the vibration frequency of the rotating device, control the vibration time and move the vibration system, the rotating device is linked with the vibration system to provide power for the vibration system. The vibration system, which is the essential part, is shown on the left side of Figure 1b. The realisation of the vibration compaction function mainly depends on the centrifugal force in the vertical direction generated by the centrifugal action of the eccentric block, and the centrifugal force of the eccentric block comes from the power generated by the high-speed operation of the motor. Theoretically, if the two sets of eccentric blocks rotate in opposite directions at the same speed, it can be considered that the centrifugal force component of the eccentric blocks in the horizontal direction cancels out to zero, and the pressed material is vibrationally compacted only by the vertical component of the centrifugal force.

The vibration parameters of the specimen are the core components of the VVTE, which are shown in Table 5.

(2) Specimen moulding

First, we placed the graded crushed stone into the test mould, lowered the vibration hammer until it was in contact with the specimen surface and set the vibration time to 100 s to determine the maximum dry density and optimal moisture content in the room. Second, according to the corresponding maximum dry density and optimal moisture content, we determined the vibration time to ascertain the compaction coefficient. When the compaction coefficient was 96, 98 and 100%, the corresponding vibration time was 70, 80 and 90 s. After vibration compaction, the moulded specimens were demoulded and cured for 3 to 90 days under standard curing conditions (temperature: 20 ± 2 °C, humidity ≥ 95%).

### 2.4. Test Methods

(1) Unconfined compressive strength

Under the configured moisture content condition, the VVCM was used to form the cement-graded crushed stone specimens with different compaction coefficients, the size of the specimen was *Φ* 200 × *h* 160 mm, and after soaking five groups of parallel specimens in water for one day and night, an unconfined compressive strength test was conducted on the pavement material strength metre specified in the specification, the loading rate is 1 mm/min [32].

The test results are calculated according to Equation (1):(1)$$R_{c}=\frac{F}{S}$$

Here, *R**_c_* is the unconfined compressive strength (MPa), *F* is the axial pressure (k N) and *S* is the area of the top surface of the specification (m^2^).

(2) Splitting strength

Under the configured moisture content condition, the VVCM was used to form the cement-graded crushed stone specimens with different compaction coefficients, the size of the specimen was *Φ* 200 × *h* 160 mm, and after soaking five groups of parallel specimens in water for one day and night, the splitting strength test was conducted using the universal testing machine specified in the standard, the loading rate is 1 mm/min [32].

The test results are calculated according to Equation (2):(2)$$R_{i}=\frac{2P}{\pi dh}\left(\sin2\alpha -\frac{a}{d}\right)$$

Here, *R_i_* is the splitting strength of the specimen (MPa), *d* is the diameter of the specimen (mm), *h* is the height of the specimen after immersion (mm), *P* is the maximum pressure (N) at the time of specimen failure, a is the width of the layering (mm) and *α* is the central angle corresponding to semi-layering (°).

(3) Resilience modulus

Under the configured moisture content condition, the VVCM was used to form the cement-graded crushed stone specimens with different compaction coefficients, the size of the specimen was *Φ* 200 × *h* 160 mm, and after soaking five groups of parallel specimens in water for one day and night, the universal testing machine specified in the standard to calculate the modulus of resilience, the loading rate is 1 mm/min [32].

The modulus of resilience under each load is calculated according to Equation (3).
(3)$$E_{e}=\frac{\pi pD}{4l}\left(1-\mu^{2}\right)$$

Here, *E_e_* is the resilience modulus (kPa); *p* is the unit pressure (kPa) on the bearing plate; *D* is the bearing plate diameter (mm); *l* is the resilience deformation corresponding to unit pressure (mm); μ is the Poisson’s ratio of the graded crushed stone.

## 3. Results and Discussion

### 3.1. Influence of Curing Times on Mechanical Properties

Figure 2 shows the relationship between unconfined compressive strength (*R_c_*) and cement-graded crushed stone curing times. Cement dosages of P_s_ = 3, 4 and 5% were used for the comparative study.

Figure 2 shows that regardless of the gradation type, the *R_c_* of the cement-graded crushed stone increased with curing time. During the first 14 days, the *R_c_* increased rapidly. The strength after 7 days was approximately 60% of the ultimate strength. The *R_c_* increased rate gradually lessened after 28 days to approximately 83% of the ultimate strength. At 90 days, the strength became approximately 95% of the ultimate strength. After this time, the *R_c_* curve approached a horizontal line, and the vertical intercept corresponding to this horizontal line was set as the ultimate unconfined compressive strength. Given the same age, the unconfined compressive strength of VGM-30 > GF1 > GF2.

The relationship between the growth law of splitting strength (*R_i_*) and cement-graded crushed stone curing time is shown in Figure 3.

Figure 3 shows that the *R_i_* growth law of cement-graded crushed stone of several types was similar to that of unconfined compressive strength, and both increase with the increase of curing time. *R_i_* had the fastest increase during the first 14 days. The strength at 7 days was approximately 50% of the ultimate strength. After 28 days, the increase in *R_i_* gradually slowed to 75% of the ultimate strength. At 90 days, the strength was approximately 90%. After this time, the *R_i_* curve approached a horizontal line, and the vertical intercept corresponding to this horizontal line was set as the ultimate splitting strength. At the same age, the splitting strength of VGM-30 > GF1 > GF2.

Figure 4 shows the relationship between the growth law and curing times of the resilience modulus (*E_e_*) of cement-graded crushed stone.

Figure 4 shows that the growth law of the resilience modulus of several types of cement-graded crushed stone was similar to the unconfined compressive strength, and both increase with longer curing times. *E_e_* increased the fastest before 14 days, and the 7-day modulus was about 54% of the ultimate modulus. After 28 days, the growth of *E_e_* gradually slowed down to about 83% of the ultimate modulus. At 90 days, the modulus was about 95% of the ultimate modulus. After 90 days, the *E_e_* curve approached a horizontal line, and we set the vertical intercept corresponding to this horizontal line as the ultimate modulus of resilience. At the same age, the resilience modulus of VGM-30 > GF2 > GF1.

In Figure 2, Figure 3 and Figure 4, the curing times had similar effects on the unconfined compressive strength, splitting strength and resilience modulus of the cement-graded crushed stone, and they all increased with increased age, with 80% of the ultimate strength being formed within the first 28 days. This result may have been due to the fact that from days 0 to 28, the hydration reaction between cement and water was violent. The hydration, hardening and coagulation reactions of cement continued to generate hydration products rapidly, and as the bonding effect of hydration products on aggregates increases, the integrity of the material improves, resulting in a rapid increase in strength. After 28 days, a large amount of gel content had been generated in the hydration reaction, the cement clinker was gradually consumed, and the rapid period of strength growth had passed, meaning the growth rate slowed down. After 90 days, which was basically the end of the reaction, the strength growth tended to stabilise, reaching the ultimate strength of cement-graded crushed stone.

### 3.2. Influence of Cement Dosage on Mechanical Properties

The highest unconfined compressive strength, splitting strength and resilience modulus of two general gradations, GF1-M and GF2-D, were selected for comparative analysis with VGM-30 and the representative curing times of 7, 28 and 90 days.

The relationship between the growth law for the unconfined compressive strength of cement-graded crushed stone and cement dosage is shown in Figure 5.

Figure 5 shows that the unconfined compressive strength of the specimen increased with the increases in cement dosage. When the cement dosage was increased by 1%, the strengths of GF1-M, GF2-D and VGM-30 graded crushed stone increased by 10, 9 and 7%, respectively. The growth rate of unconfined compressive strength during the 7–28-day period was much greater than during the 28–90-day period, with 83% of the ultimate unconfined compressive strength being formed before 28 days, demonstrating that the unconfined compressive strength of the cement-graded crushed stone was completely formed within the first 30 days. Under the same cement dosage, the unconfined compressive strength of VGM-30 > GF1 > GF2.

The relationship between the growth law for the splitting strength of cement-graded crushed stone and the cement dosage is shown in Figure 6.

Figure 6 shows that as the cement dosage increased by 1%, the splitting strength of the cement-graded crushed stones GF1-M, GF2-D and VGM-30 increased by 6, 6 and 5%, respectively. Under the same gradation, the splitting strength of the cement-graded crushed stone increased with the increase of cement dosage. Therefore, the appropriate cement dosage can significantly increase the splitting strength of cement-graded crushed stone. During the 7–28-day period, the growth rate of splitting strength is greater than during the 28–90-day period, and 75% of the ultimate splitting strength was formed before 28 days, demonstrating that the splitting strength of the cement-graded crushed stone was completely formed within the first 30 days. Under the same cement dosage, the splitting strength of VGM-30 > GF1 > GF2.

The relationship between the growth law for the resilient modulus of cement-graded crushed stone and cement dosage is shown in Figure 7.

Figure 7 shows that when the cement dosage was increased by 1%, the resilient moduli of GF1-M, GF2-D and VGM-30 graded crushed stone increased by 13%, 15% and 14%, respectively. Under the same gradation, the resilient modulus increased with the increase of cement dosage. Therefore, the resilient modulus of cement-graded crushed stone can be significantly increased by using the appropriate cement dosage. During the 7–28-day period, the growth rate of resilient modulus was much greater than during the 28–90-day period, and 83% of the ultimate resilient modulus was formed before day 28, which shows that the resilient modulus of cement-graded crushed stone was formed within the first 30 days. Under the same cement dosage, the resilient modulus of VGM-30 > GF2 > GF1.

Figure 5, Figure 6 and Figure 7 demonstrate that the effects of cement dosage on the unconfined compressive strength, splitting strength and resilient modulus of cement-graded crushed stone were similar, increasing with the increase of cement dosage and forming 80% of the ultimate strength within the first 28 days. This result may be due to the hydration reaction of cement-graded crushed stone becoming more intense with the increase of cement dosage and the production of more hydration products with increasing cementation ability. These primarily play the role of bonding and filling in cement-graded crushed stone. Therefore, the aggregate voids were gradually bonded and filled by the hydration products, resulting in the improvement of density and cohesion in the specimen. The specimen’s mechanical properties also increased with the increase of cement dosage. With the increase in curing time, the cement and water were gradually consumed, and the reaction is gradually ended; therefore, the strength and modulus growth rate of cement-graded crushed stone decreased gradually after 28 days. Although high-dosage cement can improve the mechanical properties of cement-graded crushed stone to a certain extent, the economy decreases and cement-graded crushed stone may produce shrinkage cracking, resulting in decreased performance.

### 3.3. Influence of Gradation on Mechanical Properties

The highest unconfined compressive strength, splitting strength and resilience modulus of two general gradations, GF1-M and GF2-D, were selected along with VGM-30 for comparative analysis using the representative P_s_ = 4% cement dosage.

Figure 8 presents the calculated results of the unconfined compressive strength ratio (R_1_) of VGM-30, GF1-M and GF2-D under the same conditions.

VGM-30 gradation can significantly improve the unconfined compressive strength of cement-graded crushed stone. Figure 8 shows that the unconfined compressive strength of VGM-30 gradation is always higher than that of GF1 and GF2. With the increase in curing time, the ratio gradually decreased and became stable after 14 days. However, the average value was greater than 1.1, which shows that the unconfined compressive strength of VGM-30 can be increased by 20% compared with the standard cement-graded crushed stone.

Figure 9 presents the calculated results of the splitting strength ratio (R_2_) of VGM-30, GF1-M and GF2-D under the same conditions.

Figure 9 shows that the ratio of the splitting strength between VGM-30 and GF1 and GF2 was above 1.1, indicating that the splitting strength of graded VGM-30 is always higher than that of general graded crushed stones, indicating that compared to the splitting strength of GF1 and GF2 gradations, the splitting strength of VGM-30 gradation increases faster, and its fatigue resistance is also superior.

Figure 10 presents the calculated results of the resilient modulus ratio (R_3_) of VGM-30, GF1-M and GF2-D under the same conditions.

Figure 10 shows that the resilience modulus ratio of VGM-30 to GF1 and GF2 is greater than 1.1, indicating that the resilience modulus of VGM-30 gradation is always higher than GF1 and GF2. Further, the ratio of GF1-M and GF2-D decreased with the increase in curing time.

Figure 8, Figure 9 and Figure 10 demonstrate that the unconfined compressive strength ratio and resilience modulus ratio of VGM-30 to GF1 and GF2 decreased with curing time, while the splitting strength ratio increased with curing time. The decrease in the unconfined compressive strength ratio and resilience modulus ratio with curing time may be attributed to the hydration products in the cement-graded crushed stone not having formed sufficient strength. Their strength is primarily derived from the embedding force between the skeleton of the graded crushed stone. VGM-30 belongs to the strong interlocked skeleton density type, meaning the embedding force between aggregates is higher, and the inner friction is relatively large. Compared to GF1 and GF2, the unconfined compressive strength and resilience modulus increased in the early stage, while the ratio of VGM-30 to GF1 and GF2 gradation gradually decreased with the increase in curing time. The increase in the ratio of splitting strength to curing time could have been due to GF1-M and GF2-D having skeleton-void structures. Although the skeleton was formed between the aggregates, more voids and a larger space allow for cracks. The reinforcement effect of coarse aggregates is lower than that of VGM-30, while the splitting strength of VGM-30 splitting strength increases faster than GF1-M and GF2-D under the action of coarse aggregate reinforcement and cement gelation, meaning the ratio of VGM-30 to GF1-M and GF2-D gradually increases.

### 3.4. Influence of Compaction Coefficient on Mechanical Properties

The highest unconfined compressive strength, splitting strength and resilience modulus of two general gradations, GF1-M and GF2-D, were selected for comparative analysis with VGM-30, at representative 7, 28, and 90-day curing times with a P_s_ = 4% cement dosage.

Figure 11 shows the relationship between the unconfined compressive strength and the compaction coefficient of the cement-graded crushed stone.

Figure 11 shows that the unconfined compressive strength of the cement-graded crushed stone increased with the increase of the compaction coefficient. The compaction coefficient increases by 1%, the unconfined compressive strength of GF1-M, GF2-D and VGM-30 graded crushed stone increased by 11, 12 and 5% on average, respectively.

Figure 12 shows the relationship between the splitting strength and the compaction coefficient of cement-graded crushed stone.

Figure 12 shows that the splitting strength of cement-graded crushed stone increased with the increase of the compaction coefficient. When the compaction coefficient increased by 1%, the splitting strength of GF1-M, GF2-D and VGM-30 graded crushed stone increased by 6, 6 and 6% on average, respectively.

Figure 13 shows the relationship between the resilient modulus and the compaction coefficient of the cement-graded crushed stone.

Figure 13 shows that the resilient modulus of the cement-graded crushed stone increased with the increase of compaction coefficient. When the compaction coefficient increased by 1%, the resilient modulus of GF1-M, GF2-D and VGM-30 graded crushed stone increased by 4, 4 and 4% on average, respectively.

Figure 11, Figure 12 and Figure 13 demonstrate that the unconfined compressive strength, splitting strength and resilience modulus of cement-graded crushed stone all increased with the increase of the compaction coefficient. This result may be due to the increase of the compaction coefficient, which can reduce the voids between aggregates and improve the specimen density, improving the overall strength and stability. With the increase of the compaction coefficient, the influence of gradation type on the mechanical properties of graded crushed stone gradually decreases. Therefore, increasing the compaction coefficient can significantly improve the mechanical properties of cement-graded crushed stone.

## 4. Strength Growth Equation

### 4.1. Strength Formation Mechanism

In the process of cement stabilisation of graded crushed stone, a variety of physical, chemical and physicochemical interactions occur between cement, water and crushed stone, which is the key to the gradual formation of mechanical strength. The initial strength (*R*_0_) of cement stabilised with graded crushed stone is created by physical and partial physicochemical interactions. With the increase in curing times, the cement hydration, condensation, and hardening reactions continue to proceed, and the mechanical strength also increases. With the gradual consumption of cement clinker, the growth rate of its mechanical strength also gradually decreases. When the consumption is completed, its mechanical strength stabilises and reaches the ultimate strength of cement-graded crushed stone (*R*_∞_) [33].

Assuming a strength growth equation for cement-graded crushed stone to establish a strength growth model, the equation must satisfy the following three boundary conditions.
(4)$$T=0,\;R_{T}=R_{0}\;T=\infty ,\;R_{T}=R_{\infty }\;R_{0}<R_{\infty }$$

Here, *T* is the curing time (day), *R_T_* is the strength (MPa) of cement-graded crushed stone with curing times *T* days, *R*_0_ is the initial strength (MPa) of cement-graded crushed stone with curing time 0 days and *R*_∞_ is the ultimate strength of cement-graded crushed stone (MPa).

The mechanical strength growth equation is shown in Equation (5).
(5)$$R_{T}=R_{\infty }-\frac{R_{\infty }-R_{0}}{\xi T+1}$$

Here, *ξ* is the strength increase coefficient.

### 4.2. Prediction Model of Unconfined Compressive Strength

The strength growth equation of cement-graded crushed stone can be obtained by fitting the test data to Equation (5) according to the gradation selected above. Its related parameters are shown in Table 6. In the table, *α*, *R_c_*_0_ and *R_c_*_∞_ refer to the strength growth coefficient, initial strength and ultimate strength of cement-graded crushed stone, respectively, and *R*^2^ is the correlation coefficient.

It is assumed that *R_cT_*/*R_c_*_∞_ conforms to the power function of growth with curing time, as shown in Equation (6).
(6)$$\frac{R_{cT}}{R_{c\infty }}=A_{c}\cdot \ln^{B_{c}}\left(T+1\right)+\frac{R_{c0}}{R_{c\infty }}\;\;\;T\leq 180d$$

Here, *A_c_* and *B_c_* are regression coefficients, and the results are shown in Table 7.

The *R_cT_*/*R_c_*~*T* fitting equation of cement-graded crushed stone can be expressed as:(7)$$\frac{R_{cT}}{R_{c\infty }}=0.3\ln^{0.69}\left(T+1\right)+k_{1}$$

According to Equation (7), to establish a prediction model for the unconfined compressive strength of cement-graded crushed stone, see Equation (8).
(8)$$R_{cT}=[0.5\ln^{0.69}\left(T+1\right)+1.67k_{1}]\cdot R_{c7}$$

Here, *R_cT_* and *R_c_*_7_ are the unconfined compressive strength (MPa) of cement-graded crushed stone with curing time of *T* days and 7 days respectively, *k*_1_ is the coefficient of gradation type (take 0.15).

The ratio of the strength *R_cT_* to the ultimate strength *R_c_*_∞_ of the cement-graded crushed stone at various curing times is shown in Figure 14.

Figure 14 shows that although different cement dosages were used in the three gradations, the growth curve of the strength of the cement-graded crushed stone was normalised.

### 4.3. Prediction Model of Splitting Strength

Equation (5) was used to fit the test data to obtain the splitting strength growth equation of cement-graded crushed stone. The representative grading-related parameters selected above are shown in Table 8. In the table, *β*, *R_i_*_0_ and *R_i_* refer to the strength growth coefficient, initial strength and ultimate strength of cement-graded crushed stone, respectively, and *R*^2^ is the correlation coefficient.

If the growth of *R_iT_*/*R_i_*_∞_~*T* with curing time conforms to the power function, see Equation (9).
(9)$$\frac{R_{iT}}{R_{i\infty }}=A_{i}\cdot \ln^{B_{i}}\left(T+1\right)\;\;\;T\leq 180d$$

Here, *A_i_* and *B_i_* are the regression coefficient and the results are shown in Table 9.

The fitting equation of *R_iT_*/*R_i_*_∞_~*T* for cement-graded crushed stone can be expressed as:(10)$$\frac{R_{iT}}{R_{i\infty }}=0.35\ln^{0.63}\left(T+1\right)$$

According to Equation (10), the splitting strength prediction model of cement-graded crushed stone is established, as shown in Equation (11).
(11)$$R_{iT}=0.70\ln^{0.63}\left(T+1\right)\cdot R_{i7}$$

Here, *R_iT_* and *R_i_*_7_ are the splitting strengths (MPa) of cement-graded crushed stone with curing times *T* days and 7 days, respectively.

The ratio of strength *R_iT_* to ultimate strength *R_i∞_* of graded crushed stone at various curing times is shown in Figure 15.

Figure 15 shows that the strength growth curve of the cement-graded crushed stone with different cement dosages has been normalised.

### 4.4. Prediction Model of Resilience Modulus

Equation (5) was used to fit the test data to obtain the growth equation regarding the graded crushed stone’s resilience modulus. The relevant parameters selected above are shown in Table 10. In the table, γ, *E_e_*_0_ and *E_e∞_* refer to the modulus growth coefficient, initial modulus and ultimate modulus of cement-graded crushed stone, respectively, while *R*^2^ is the correlation coefficient.

If the growth of *E_eT_*/*E_e_*_∞_ conforms to the power function, see Equation (12).
(12)$$\frac{E_{eT}}{E_{e\infty }}=A_{e}\cdot \ln^{B_{e}}\left(T+1\right)+\frac{E_{e0}}{E_{e\infty }}\;\;\;T\leq 180d$$

Here, *A_e_* and *B_e_* are the regression coefficients. The three gradations are compared and analysed, and the results are shown in Table 11.

The *E_eT_*/*E_e∞_*~*T* fitting equation of cement-graded crushed stone is shown in Equation (13).
(13)$$\frac{E_{eT}}{E_{e\infty }}=0.23\ln^{0.86}\left(T+1\right)+k_{2}$$

Based on Equation (13), the prediction model of resilience modulus for cement-graded crushed stone is established in Equation (14).
(14)$$E_{eT}=[0.43\ln^{0.86}\left(T+1\right)+1.85k_{2}]\cdot E_{e7}$$

Here, *E_eT_* and *E_e_*_7_ are the resilience modulus (MPa) of cement-graded crushed stone with curing times *T* days and 7 days, respectively, and *k*_2_ is the coefficient of the gradation type (take 0.15).

The ratio of *E_eT_* and ultimate modulus *E_e∞_* of cement-graded crushed stone at each curing time and the relationship between *E_eT_*/*E_e∞_*~*T* are shown in Figure 16.

Figure 16 shows that the strength growth curve of cement-graded crushed stone with different cement dosages has been normalised.

### 4.5. Reliability Evaluation of the Mechanical Strength Growth Equation

A representative specimen of cement-graded crushed stone with a cement dosage of 4%, compaction coefficient of 98% and VGM-30 gradation was selected for unconfined compressive strength, splitting strength and resilience modulus tests. The measured values and predicted values were compared, and the results are shown in Table 12.

Table 12 shows that the predicted value of unconfined compressive strength of the cement-graded crushed stone had a maximum error of 7.9% and an average error of 3.6%, while the maximum error between the predicted value of splitting strength and the measured value was 10.6% and an average error was 5.1%, and the maximum error between the predicted value of the resilience modulus and the measured value was 8.4%, while the average error is 3.8%. The results show that the mechanical strength prediction model established in this study can accurately predict the growth law of the mechanical strength of cement-graded crushed stone.

## 5. Conclusions

In this paper, the strength growth model of mechanical properties was established, and the influence of curing times, cement dosage, gradation type and compaction coefficient on the mechanical properties of cement-graded crushed stone was studied. The following conclusions were drawn:(1)A strength growth model of cement-graded crushed stone is proposed, and strength prediction models of unconfined compressive strength, splitting strength and resilience modulus of cement-graded crushed stone are proposed based on this model. These can accurately predict the growth trends of mechanical properties in cement-graded crushed stone.(2)VGM-30 grading can significantly improve the unconfined compressive strength, splitting strength and resilience modulus of cement-graded crushed stone compared with standard cement-graded crushed stone. The overall unconfined compressive strength, splitting strength and resilience modulus can be increased by an average of 20, 20 and 17%, respectively.(3)The mechanical strength and modulus of the graded crushed stone increase with the increase of cement dose. For every 1% increase in cement dosage, the unconfined compressive strength can be increased by at least 7%, the splitting strength can be increased by at least 5% and the resilience modulus can be increased by at least 12%.(4)The mechanical properties of cement-graded crushed stone can be significantly improved by increasing the compaction coefficient, the compaction coefficient is increased by 1%, and the unconfined compressive strength can be increased by at least 5%, the splitting strength can be increased by at least 6% and the resilience modulus can be increased by at least 4%.(5)The three mechanical properties of cement-graded crushed stone increased with an increase in curing time, and the growth laws were similar, with all increasing fastest during the first 28 days and then decreasing after this time. After 90 days, the increase in strength tended to stabilise.

## Figures and Tables

**Figure 1 materials-15-02132-f001:**
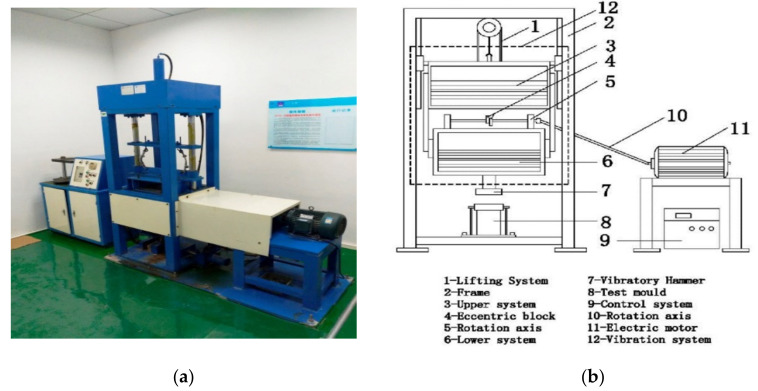
VVTE structure and working principle. (**a**) Photograph of VVTE, (**b**) VVTE schematic.

**Figure 2 materials-15-02132-f002:**
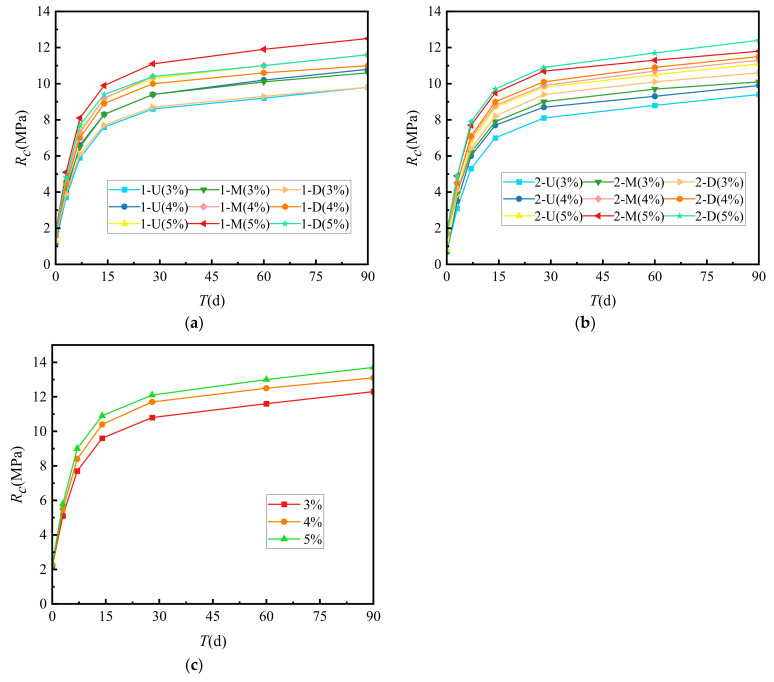
Growth law of unconfined compressive strength. (**a**) GF1, (**b**) GF2, (**c**) VGM-3.

**Figure 3 materials-15-02132-f003:**
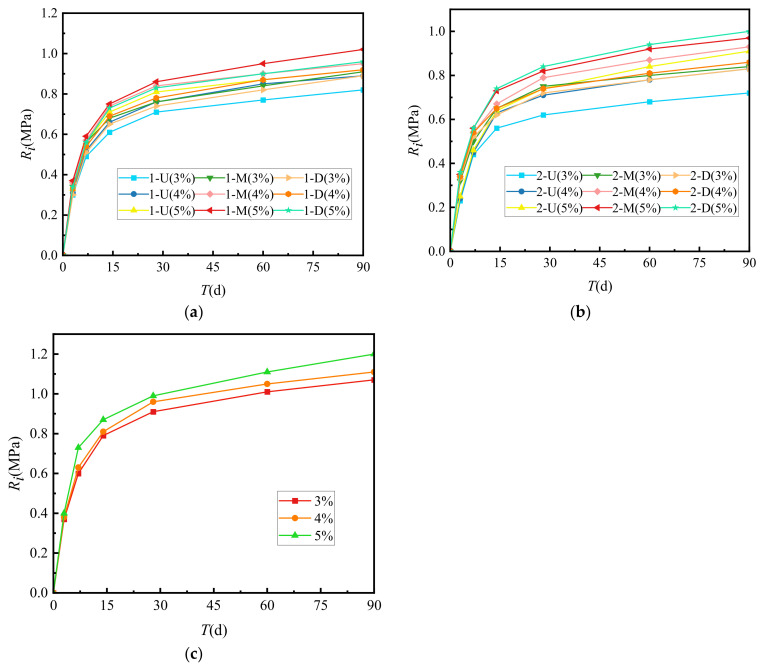
Growth law of splitting strength. (**a**) GF1, (**b**) GF2, (**c**) VGM-30.

**Figure 4 materials-15-02132-f004:**
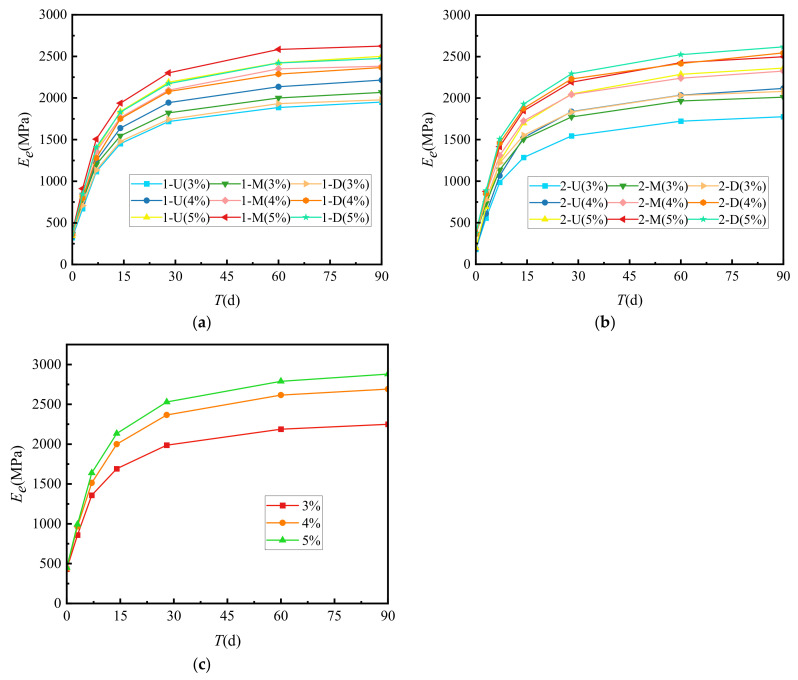
Growth law of resilience modulus. (**a**) GF1, (**b**) GF2, (**c**) VGM-30.

**Figure 5 materials-15-02132-f005:**
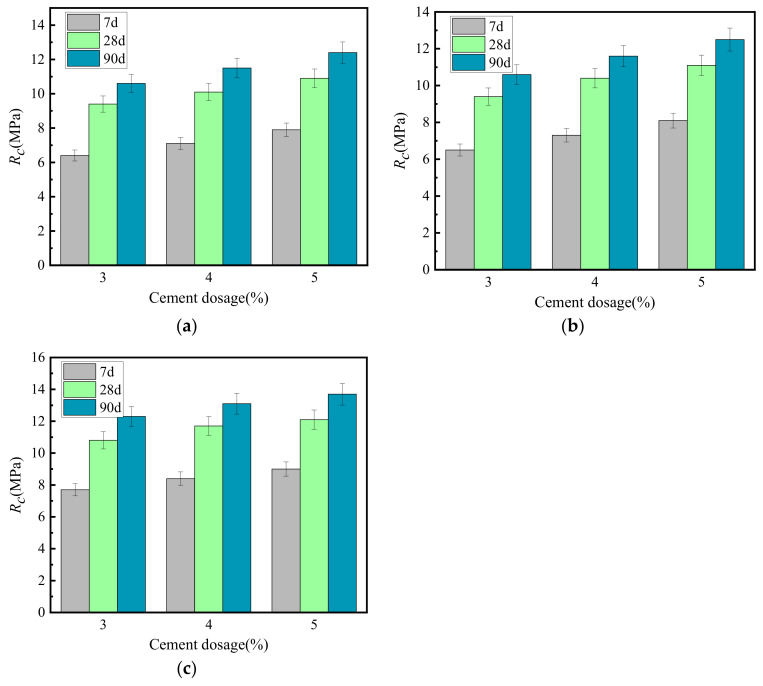
Growth law of unconfined compressive strength under different cement dosages. (**a**) GF1, (**b**) GF2, (**c**) VGM-30.

**Figure 6 materials-15-02132-f006:**
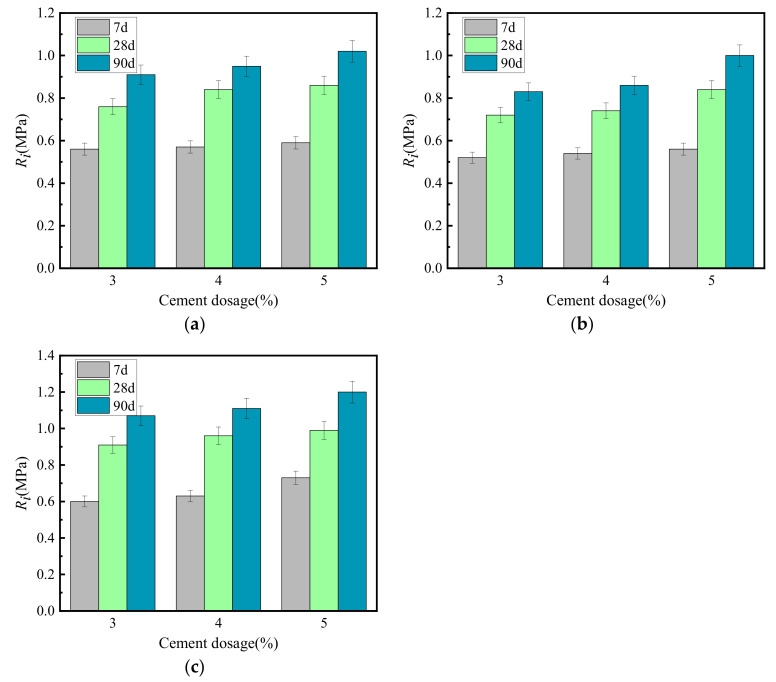
Growth law of splitting strength under different cement dosages. (**a**) GF1, (**b**) GF2, (**c**) VGM-30.

**Figure 7 materials-15-02132-f007:**
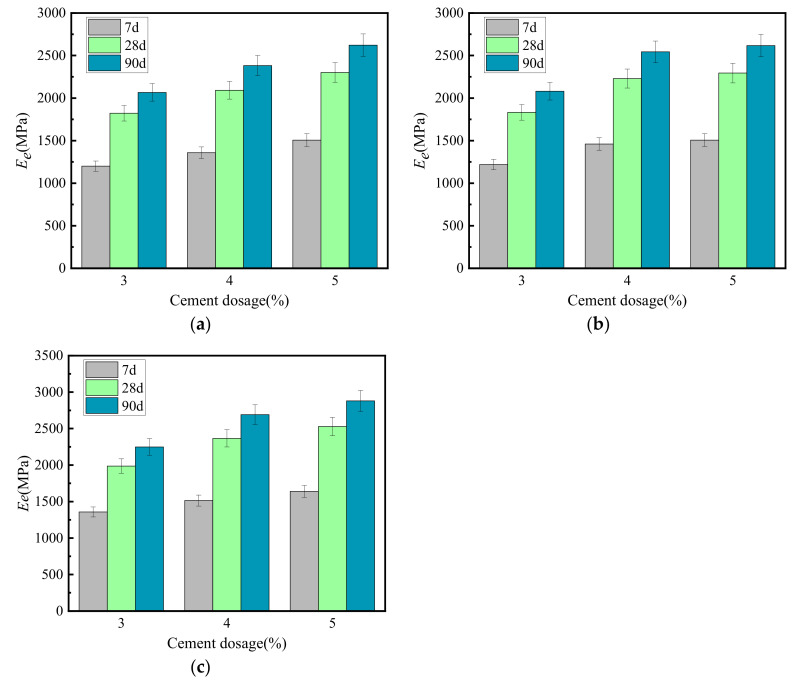
Growth law of resilient modulus under different cement dosage. (**a**) GF1, (**b**) GF2, (**c**) VGM-30.

**Figure 8 materials-15-02132-f008:**
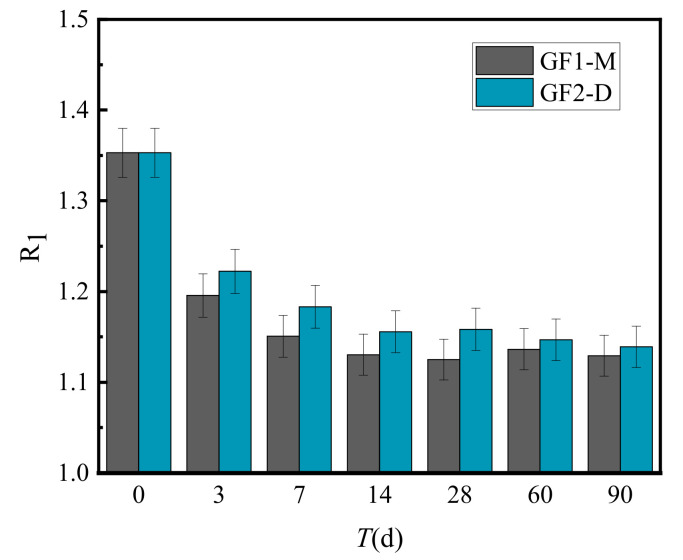
Unconfined compression strength ratio of VGM-30 and general gradations.

**Figure 9 materials-15-02132-f009:**
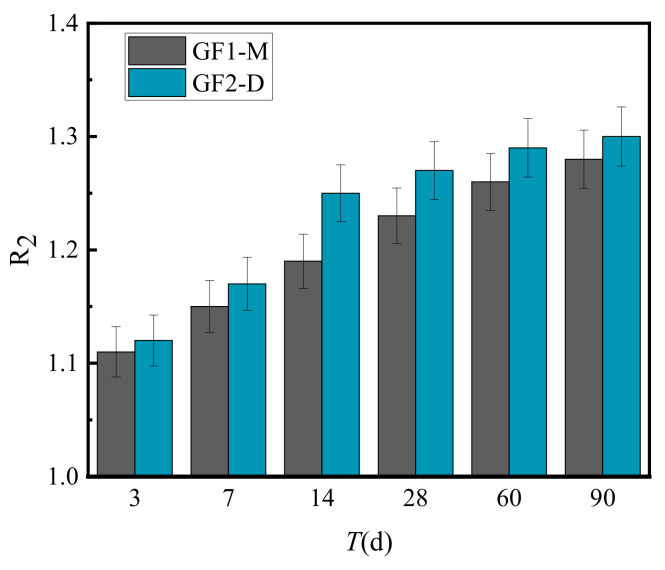
Splitting strength ratio of VGM-30 and general gradations.

**Figure 10 materials-15-02132-f010:**
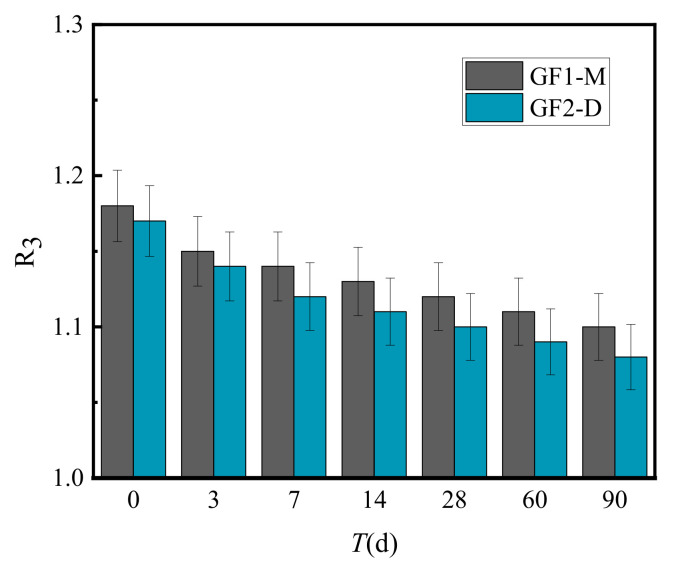
Resilience modulus ratio of VGM-30 and general gradations.

**Figure 11 materials-15-02132-f011:**
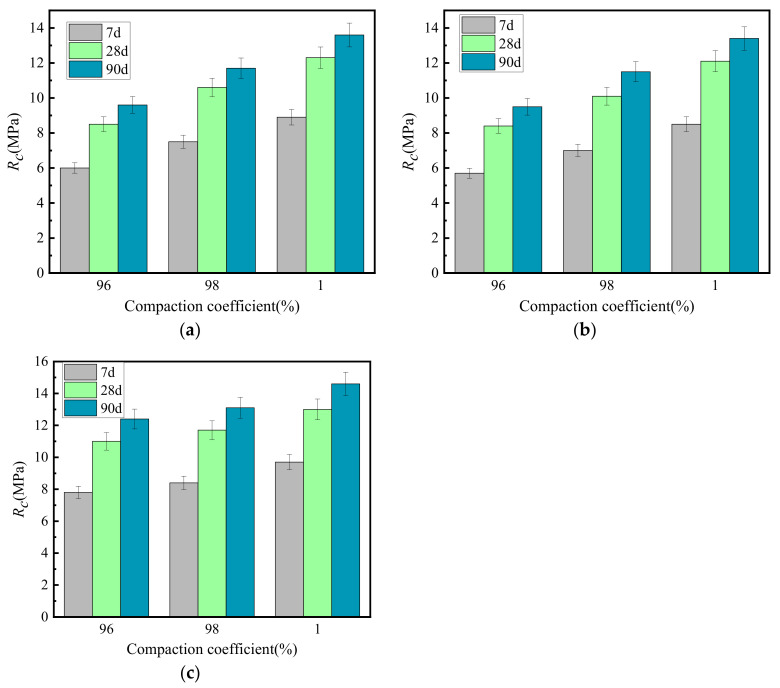
Growth law of unconfined compressive strength under different compaction coefficients. (**a**) GF1, (**b**) GF2, (**c**) VGM-30.

**Figure 12 materials-15-02132-f012:**
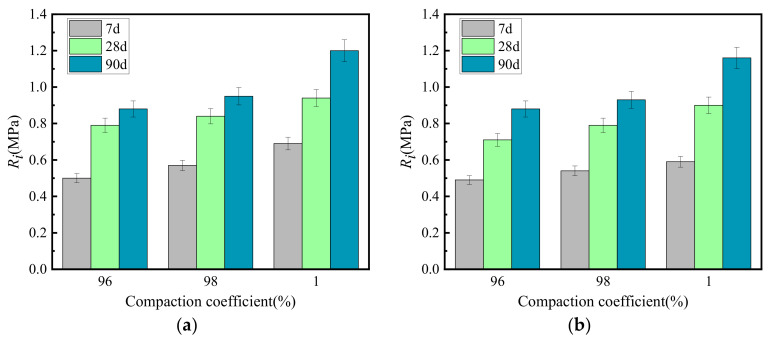
Growth law of splitting strength under different compaction coefficients. (**a**) GF1, (**b**) GF2, (**c**) VGM-30.

**Figure 13 materials-15-02132-f013:**
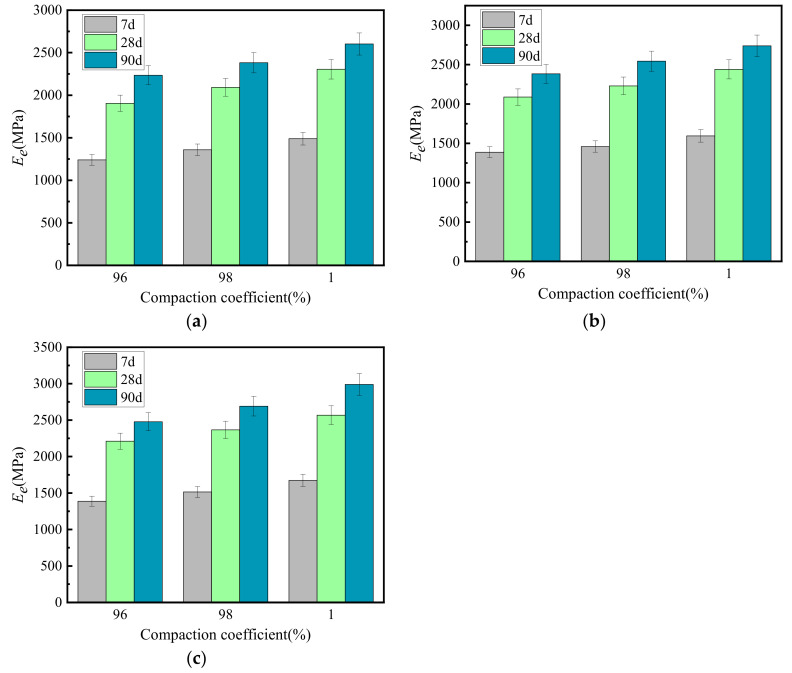
Growth law of resilient modulus under different compaction coefficients. (**a**) GF1, (**b**) GF2, (**c**) VGM-30.

**Figure 14 materials-15-02132-f014:**
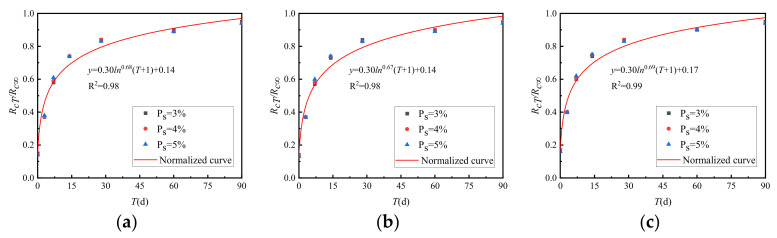
Growth law of *R_cT_*/*R_c_*_∞_ between general grading and VGM-30. (**a**) GF1, (**b**) GF2, (**c**) VGM-30.

**Figure 15 materials-15-02132-f015:**
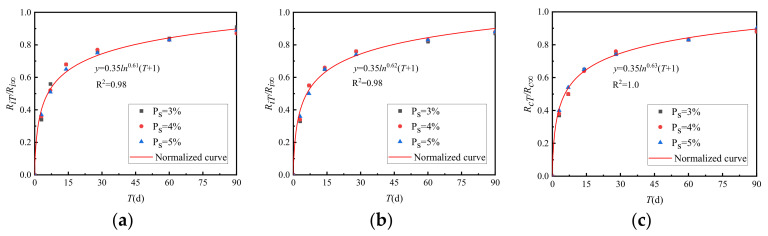
Growth law of *R_iT_*/*R_i∞_* between general grading and VGM-30. (**a**) GF1, (**b**) GF2, (**c**) VGM-30.

**Figure 16 materials-15-02132-f016:**
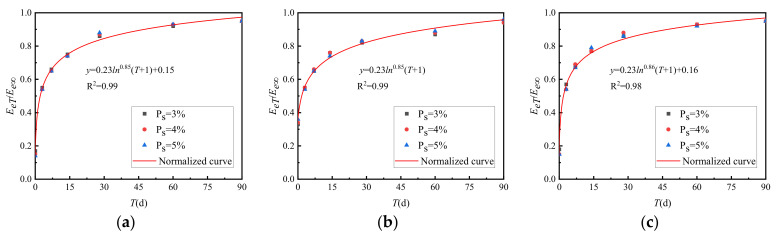
Growth law of *E_eT_*/*E_e∞_* between general grading and VGM-30. (**a**) GF1, (**b**) GF2, (**c**) VGM-30.

**Table 1 materials-15-02132-t001:** Main physical properties of graded gravel.

Index	Measured Values	Standard
Needle content (%)	8.7	≤18
Crush value (%)	9.0	≤28
Los Angeles attrition rate (%)	24.1	≤30
Plastic index	5.7	≤4
Porosity (%)	25	<28
Bibulous rate (%)	2	<4

**Table 2 materials-15-02132-t002:** Technical properties of cement.

Index	Density (g/cm^3^)	Stability	Water for Standard Consistency (%)	Specific Surface Area (m^2^/kg)	Initial Setting Time (min)	Final Setting Time (min)
Test results	3.07	qualified	27.3	342	292	388

**Table 3 materials-15-02132-t003:** Aggregate gradation.

Gradation Type		Percentage (Mass%) Passing through the Sieve Aperture (mm)								
		53	37.5	31.5	19	9.5	4.75	2.36	0.6	0.075
GF1	Upper	100	100	-	90	-	65	50	30	10
	Median	100	97.5	-	75	-	47.5	35	20	6
	Down	100	95	-	60	-	30	20	10	2
GF2	Upper	-	100	100	90	-	65	50	30	10
	Median	-	100	97.5	75	-	47.5	35	20	6
	Down	-	100	95	60	-	30	20	10	2
VGM-30		-	-	100	65	51	30	24	15	-

**Table 4 materials-15-02132-t004:** Optimal moisture content and maximum dry density.

Gradation Type	*P*_s_ (%)	GF1		GF2		VGM-30	
		*W* (%)	*ρ*_dmax_ (g/cm^3^)	*W* (%)	*ρ*_dmax_ (g/cm^3^)	*W* (%)	*ρ*_dmax_ (g/cm^3^)
U	3	4.4	2.471	4.5	2.469	3.8	2.512
	4	4.8	2.480	4.9	2.478	3.9	2.529
	5	5.1	2.488	5.3	2.485	4.1	2.547
M	3	4.1	2.487	4.3	2.484	-	-
	4	4.3	2.504	4.5	2.500	-	-
	5	4.6	2.519	4.7	2.516	-	-
D	3	3.9	2.467	4.0	2.483	-	-
	4	4.1	2.486	4.2	2.501	-	-
	5	4.3	2.505	4.3	2.517	-	-

**Table 5 materials-15-02132-t005:** VVTE vibration parameter configuration.

Work Frequency (Hz)	Work Amplitude (mm)	Work Weight (kg)		
		Upper System Weigh	Lower System Weight	Total Weight
30	1.2	120	180	300

**Table 6 materials-15-02132-t006:** Unconfined compressive strength growth equation parameters.

Gradation Type	P_s_ (%)	*α*	*R_c_*_0_ (MPa)	*R_c_*_∞_ (MPa)	*R* ^2^
GF1-M	3	0.15	1.7	11.2	1.00
	4	0.15	1.7	12.4	0.99
	5	0.15	1.8	13.3	0.99
GF2-D	3	0.15	1.6	11.2	1.00
	4	0.15	1.7	12.2	0.99
	5	0.15	1.7	13.1	1.00
VGM30	3	0.15	2.2	12.9	0.99
	4	0.15	2.3	13.9	0.99
	5	0.15	2.3	14.5	1.00

**Table 7 materials-15-02132-t007:** Fitting regression coefficients of *R_cT_*/*R_c_*_∞_~*T*.

Gradation Type	*A* _c_	*B* _c_	*R* ^2^
GF1-M	0.30	0.68	0.98
GF2-D	0.30	0.67	0.98
VGM-30	0.30	0.69	0.99
Average	0.30	0.69	0.99

**Table 8 materials-15-02132-t008:** Splitting strength growth equation parameters.

Gradation Type	P_s_ (%)	*β*	*R_i_*_∞_ (MPa)	*R* ^2^
GF1-M	3	0.10	1.00	0.95
	4	0.10	1.09	0.97
	5	0.10	1.15	0.98
GF2-D	3	0.10	0.95	0.97
	4	0.10	0.98	0.97
	5	0.10	1.13	0.98
VGM-30	3	0.10	1.21	0.97
	4	0.10	1.26	0.98
	5	0.10	1.34	0.99

**Table 9 materials-15-02132-t009:** Fitting regression coefficients of *R_iT_*/*R_i∞_*~*T*.

Gradation Type	*A* _i_	*B* _i_	*R* ^2^
GF1-M	0.35	0.61	0.98
GF2-D	0.35	0.61	0.98
VGM-30	0.35	0.63	1.00
Average	0.35	0.63	0.99

**Table 10 materials-15-02132-t010:** Resilience modulus growth equation parameters.

Gradation Type	P_s_ (%)	*γ*	*E_e_*_0_ (MPa)	*E_e_*_∞_ (MPa)	*R* ^2^
GF1-M	3	0.15	367	2167	1.00
	4	0.15	374	2515	0.99
	5	0.15	390	2772	0.99
GF2-D	3	0.15	361	2189	0.98
	4	0.15	368	2661	0.99
	5	0.15	376	2750	1.00
VGM-30	3	0.15	431	2362	1.00
	4	0.15	437	2825	0.99
	5	0.15	450	3024	1.00

**Table 11 materials-15-02132-t011:** Fitting regression coefficients of *E_eT_/E_e∞_*~*T*.

Gradation Type	*A* _e_	*B* _e_	*R* ^2^
GF1-M	0.23	0.85	0.99
GF2-D	0.23	0.85	0.99
VGM-30	0.23	0.86	0.98
Average	0.23	0.86	0.99

**Table 12 materials-15-02132-t012:** The comparison of predicted and measured values.

Test Items	Curing Time (Days)	Measured Values (MPa)	Predicted Values (MPa)	Error (%)
Unconfined compressive strength	7	8.25	8.90	7.9
	14	10.40	10.27	1.2
	28	11.70	11.60	0.85
	60	12.51	13.01	4.0
	90	13.18	13.73	4.2
Splitting strength	7	0.66	0.73	10.6
	14	0.83	0.87	4.8
	28	0.97	0.99	2.1
	60	1.08	1.13	4.6
	90	1.15	1.19	3.5
Resilience modulus	7	1514	1642	8.4
	14	2000	1954	2.3
	28	2366	2267	4.2
	60	2616	2616	0
	90	2691	2798	4.0

## Data Availability

Some or all data, models, or code that support the findings of this study are available from the corresponding author upon reasonable request.

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
