# Peer review of "Research on Mechanical Properties and Influencing Factors of Cement-Graded Crushed Stone Using Vertical Vibration Compaction"

_materials, 2022, doi:10.3390/ma15062132_

Round 1

Reviewer 1 Report

Thank you for explonation about my previous comments and opinion that reviewed paper is not the same as the previous published paper (Transportation Geotechnics) in terms of research objectives, serearch methods and content. 

Additional comments:

1) Please add additional information about Vertical Vibration Testing Equipment (VVTE). Was that equipment design and use as typical equipment or just adapted for the research.

2) Acc. 2.4. How many specimens were used in each test method? Five groups of specimens after soaking means just 5 specimens for VGM-30 and 5 standard cement-graded crushed stone?

Author Response

Point 1: I Please add additional information about Vertical Vibration Testing Equipment (VVTE). Was that equipment design and use as typical equipment or just adapted for the research.

Response 1: Please provide your response for Point 1. (in red)

The overall structure of the vertical vibration test equipment (VVTE) is shown in Figure 1. The equipment is composed of three parts: the control platform, the rotating device and the vibration system. The function of the vibration platform is to adjust the vibration frequency of the rotating device, control the vibration time,and move the vibration system, the rotating device is linked with the vibration system to provide power for the vibration system. As shown in Figure 1(b), the vibration system, which is the essential part, is show on the left side of Figure 1(b). The realization of the vibration compaction function mainly depends on the centrifugal force in the vertical direction generated by the centrifugal action of the eccentric block, and the centrifugal force of the eccentric block comes from the power generated by the high-speed operation of the motor. Theoretically, if the two sets of eccentric blocks rotate in opposite directions at the same speed, it can be considered that the centrifugal force component of the eccentric blocks in the horizontal direction cancels out to zero, and the pressed material is vibrationally compacted only by the vertical component of the centrifugal force.

Point 2: Acc. 2.4. How many specimens were used in each test method? Five groups of specimens after soaking means just 5 specimens for VGM-30 and 5 standard cement-graded crushed stone?et it. Alternatively, they may use the proportion approach which will make the interpretation easier.

Response 2: Please provide your response for Point 2. (in red)

Each test in each gradation adopts 5 groups of specimens, and the average value is taken as the test data. Because the dispersion of test data was found to be small during the analysis of test results, the Coefficient of Variationwas calculation and CV<10%. Therefore, 5 groups of test values were selected for each test.

Reviewer 2 Report

Research on mechanical properties and influencing factors of cement-graded crushed stone using vertical vibration compaction

This manuscript showed an attempt to investigate the mechanical performance of cement-graded crushed stone for use in the transition sections of intercity railways. It is a subject with increasing interest from the research community and, also, from the industry. The paper's quality is significant as it considered previously comments addressed by the reviewer (first submission) and thus justifies publication with no further revisions.

Author Response

Thank you very much for your review

This manuscript is a resubmission of an earlier submission. The following is a list of the peer review reports and author responses from that submission.

Round 1

Reviewer 1 Report

The reviewed paper is very close to previous published paper in Transportation Geotechnics. Available online 9 March 2021. Title: "Mechanical properties and influencing factors of vertical-vibration compacted unbound graded aggregate materials". Authors of that paper are: Changqing Deng, Yingjun Jiang, Yu Zhang, Yong Yi, Tian Tian Kejia Yuan, Jiangtao Fan. Please consider to add the clear objectives in reviewed paper to avoid plagiarism. Also methods used, materials and results should be different.

Reviewer 2 Report

Research on mechanical properties and influencing factors of cement-graded crushed stone using vertical vibration compaction

This manuscript showed an attempt to investigate the mechanical performance of cement-graded crushed stone for use in the transition sections of intercity railways. It is a subject with increasing interest from the research community and, also, from the industry. The paper's quality is significant and justifies publication after a few revisions.

  • Introduction: “ At present, some researchers have carried out studies on cement-graded macadam….etc”. The presentation of the literature review is not suitable. Please try to present in a better writing format (list key findings in the area of research and support by citing proper publications then show the gap)
  • Section 2.3: Reference the user manual of the VVTE equipment.
  • Section 2.4: missing numbering for part 1 unconfined compressive strength (only 2 for splitting strength and 3 for Resilience modulus)
  • Section 2.4:VVCM was used to form the cement-graded crushed stone specimens with different compaction coefficients, so why the two sizes? Are they both used for the unconfined compressive strength test? Same question for Splitting strength and Resilience modulus tests.
  • Section 2.4: The height of the specimen) is less than the diameter for the two sizes mentioned. Could this compromise the test as 2:1 height: diameter ratio is always recommended for the unconfined compressive test.
  • Section 2.4: details for all tests are missing (ex: test mechanism, test condition, rate of loading, number of specimens tested (reliability of results),…etc)